# Formation of Pre-PCTA/DT Intermediates from 2-Chlorothiophenol on Silica Clusters: A Quantum Mechanical Study

**DOI:** 10.3390/ijms25063485

**Published:** 2024-03-20

**Authors:** Fei Xu, Xiaotong Wang, Ying Li, Yongxia Hu, Ying Zhou, Mohammad Hassan Hadizadeh

**Affiliations:** 1Environment Research Institute, Shandong University, Qingdao 266237, China; 202113003@mail.sdu.edu.cn (X.W.); sdliying@mail.sdu.edu.cn (Y.L.); 202232994@mai.sdu.edu.cn (Y.H.); 202333014@mail.sdu.edu.cn (Y.Z.); 2Shenzhen Research Institute of Shandong University, Shenzhen 518057, China; 3Hefei National Laboratory for Physical Sciences at the Microscale, Synergetic Innovation Center of Quantum Information and Quantum Physics, University of Science and Technology of China, Hefei 230026, China

**Keywords:** 2-chlorothiophenol, silica clusters, pre-PCTA/DT formation, L–H and E–R mechanism, DFT calculations

## Abstract

Silica (SiO_2_), accounting for the main component of fly ash, plays a vital role in the heterogeneous formation of polychlorinated thianthrenes/dibenzothiophenes (PCTA/DTs) in high-temperature industrial processes. Silica clusters, as the basic units of silica, provide reasonable models to understand the general trends of complex surface reactions. Chlorothiophenols (CTPs) are the most crucial precursors for PCTA/DT formation. By employing density functional theory, this study examined the formation of 2-chlorothiophenolate from 2-CTP adsorbed on the dehydrated silica cluster ((SiO_2_)_3_) and the hydroxylated silica cluster ((SiO_2_)_3_O_2_H_4_). Additionally, this study investigated the formation of pre-PCTA/DTs, the crucial intermediates involved in PCTA/DT formation, from the coupling of two adsorbed 2-chlorothiophenolates via the Langmuir–Hinshelwood (L–H) mechanism and the coupling of adsorbed 2-chlorothiophenolate with gas-phase 2-CTP via the Eley–Rideal (E–R) mechanism on silica clusters. Moreover, the rate constants for the main elementary steps were calculated over the temperature range of 600–1200 K. Our study demonstrates that the 2-CTP is more likely to adsorb on the termination of the dehydrated silica cluster, which exhibits more effective catalysis in the formation of 2-chlorothiophenolate compared with the hydroxylated silica cluster. Moreover, the E–R mechanism mainly contributes to the formation of pre-PCTAs, whereas the L–H mechanism is prone to the formation of pre-PCDTs on dehydrated and hydroxylated silica clusters. Silica can act as a relatively mild catalyst in facilitating the heterogeneous formation of pre-PCTA/DTs from 2-CTP. This research provides new insights into the surface-mediated generation of PCTA/DTs, further providing theoretical foundations to reduce dioxin emission and establish dioxin control strategies.

## 1. Introduction

Polychlorinated thianthrenes/dibenzothiophenes (PCTA/DTs) represent structural analogues of polychlorinated dibenzo-p-dioxin/benzofurans (PCDD/Fs), in which the oxygen atoms are replaced by sulfur atoms [1]. Owing to their structural similarity, PCTA/DTs display high toxicity, long-lasting environmental persistence and physicochemical characteristics, making them classified as highly toxic persistent organic pollutants (POPs) that destroy human health and the ecosystem [2]. PCTA/DTs have been widely detected in various environmental matrices, such as soil and water sediments [3,4], pulp mill effluents [5], industrial gas samples [6], automobile exhaust [7] and petroleum refineries [8]. In some regions, higher concentrations of PCTA/DTs have been detected compared to those of PCDD/Fs [9]. Considering the high toxicity and widespread distribution of PCTA/DTs, the elucidation of the PCTA/DT formation mechanism is essential to reduce the presence of PCTA/DTs in the environment, thereby preventing potential public health threats posed by PCTA/DTs.

High relationships between concentrations of PCTA/DTs and PCDD/Fs in the environmental samples suggest similar origins and formation mechanisms of PCTA/DTs with PCDD/Fs [10,11,12]. In terms of PCTA/DT emissions, it is widely recognized that the dominant sources of PCTA/DTs in the environment are multiple combustion and industrial thermal processes, especially municipal waste incinerations [13,14,15,16]. The heterogeneous reactions in the post-flame and cool zone contribute to 70% of PCTA/DT formation [16]. Currently, the surface-mediated formation mechanism of PCTA/DTs can be divided into two categories: de novo synthesis in which residual carbon can react with halogen and sulfur elements on the surface of combustion-generated fly ash and precursor synthesis involving the catalytical coupling of chlorinated organic molecules [17,18]. Chlorothiophenols (CTPs) are important for the surface-mediated formation of PCTA/DTs because they are commonly demonstrated as crucial intermediates in de novo synthesis and predominant precursors in precursor synthesis [10,18,19,20]. Under pyrolysis or oxidative conditions, CTPs can chemically adsorb on the catalytic surface leading to the formation of chlorothiophenolates, which is the initial and crucial step in the heterogeneous formation of PCTA/DTs [21,22]. For the subsequent steps in forming the pre-PCTA/DTs, the important intermediates involved in PCTA/DT formation, the Eley–Rideal (E–R) mechanism and the Langmuir–Hinshelwood (L–H) mechanism, have been proposed [22,23]. The former refers to the reactions of surface-bound species with gaseous species, while the latter involves the coupling of two adsorbed surface species. Varieties of experimental research have proved that the L–H mechanism is responsible for the formation of PCDTs and the E–R mechanism is mainly related to the formation of PCTAs [17,22,23,24,25]. Nganai et al. [25] have elucidated differences between the L–H and E–R mechanisms, which contribute to different rate orders during the oxidation of 2-chlorophenol on the CuO/Silica surface. Typically, L–H reactions exhibit lower reaction orders compared to similar reactions in the E–R mechanism [25]. However, there are still a lot of uncertainties about the detailed heterogeneous mechanism of PCTA/DTs. A key goal is to identify the determining step in both L–H and E–R mechanisms and to determine their relative favorability. Another consideration is whether the formed PCTA/DTs remain adsorbed on the surface or are promptly released into the gas phase. Quantum chemical calculations serve as a valuable tool to bridge knowledge gaps by calculating thermodynamic and kinetic parameters for elementary steps involved in the heterogeneous formation of PCTA/DTs, thus identifying energetically favorable reaction pathways. 

During industrial thermal processes, a substantial volume of fly ash, the solid residue, is generated. Notably, fly ash exhibits remarkably catalytic capabilities in the heterogenous formation of PCTA/DTs. These catalytic effects can be attributed to the coexistence of residual organic species, particulate components (such as SiO_2_) and metal oxides (such as Cu, Al and Fe) in the fly ash [26,27,28]. Silica (SiO_2_) comprises 5−50% of fly ash and can function as a supporting substrate for transition metals [29]. Generally, varieties of transition metal compounds are regarded as the main catalytic promoters for the heterogeneous formation of dioxin [30,31,32,33,34]. However, it is reasonable to assume that silica, as the most abundant component in fly ash, plays an important role in the surface-mediated formation of PCTA/DTs. An intriguing question is whether neat silica can catalyze the formation of PCTA/DTs. Mosallanejad et al. provided experimental and theoretical evidence that PCDD/Fs can be formed from a series of chlorophenols (CPs) with the mild catalysis of neat silica, even in the absence of transition metals [35]. Fourier transform infrared (FTIR) spectroscopy by Alderman and Dellinge has detected the formation of chlorophenolates on silica surfaces, which represent crucial intermediates in the formation of PCDD/Fs [36]. Meanwhile, Pan et al. [37] corroborated the formation of chemisorbed 2-chlorophenolate on dehydrated and hydroxylated silica clusters from a theoretical aspect. Despite the fact that numerous laboratory and theoretical studies have been conducted, studies still lack a detailed formation mechanism of PCTA/DTs catalyzed by neat silica surfaces. It has been proved that employing the silica cluster model as the starting point is reasonable for investigating the complex influence of neat silica on PCTA/DT formation. Due to the presence of a significant amount of water vapor in waste incinerators and industrial systems, some silica clusters can be hydroxylated and converted to hydroxylated silica clusters. This realization has spurred us to conduct a more in-depth investigation into the heterogeneous formation of PCTA/DT from CTP precursors on neat silica clusters, focusing on the underlying mechanistic aspects.

To this end, the present contribution has employed density functional theory (DFT) to investigate the heterogeneous formation of pre-PCTA/DTs from 2-CTP with the assistance of silica clusters. In this study, the formation of 2-chlorothiophenolate via the adsorption and S-H dissociation of 2-CTP on the dehydrated and hydroxylated silica clusters has been investigated. Moreover, we further explored all possible formation routes of pre-PCTA/DT intermediates from the coupling of two adsorbed 2-chlorothiophenolates via the L–H mechanism and the coupling of adsorbed 2-chlorothiophenolate with gas-phase 2-CTP via the E–R mechanism on the dehydrated and hydroxylated silica clusters. Moreover, the vital roles of those two kinds of silica clusters in the heterogeneous formation of pre-PCTA/DTs were clarified. The rate constants for the main elementary steps were calculated over the temperature range of 600–1200 K using TST theory with Wigner tunneling correction. This research establishes a solid theoretical foundation for our comprehension of intricate mechanisms behind the heterogeneous generation of PCTA/DTs, providing in-depth insights and important input parameters for the control and prediction model of PCTA/DTs.

## 2. Results

### 2.1. Silica Clusters

Although hydroxylated silica accounts for a significant portion of silica in ambient conditions, it can easily undergo dehydroxylation reactions under high-temperature conditions [38]. In this study, Figure 1 displays two types of silica clusters, the dehydrated silica cluster ((SiO_2_)_3_) and the hydroxylated silica cluster ((SiO_2_)_3_O_2_H_4_), which have proved reasonable for investigating the formation of dioxin-like compounds [35,37]. It is also important to note that the silica cluster models employed in this work are approximations of real surfaces. These models allow us to explore the intricate influence of neat silica on PCTA/DT formation, providing valuable insights despite their simplifications. It can be seen that both the (SiO_2_)_3_ and (SiO_2_)_3_O_2_H_4_ clusters contain two rhombic Si-O rings, which have been detected in the surface or the interior of amorphous and crystalline silica [39,40]. In Figure 1a, the (SiO_2_)_3_ cluster comprises two tricoordinate silicon atoms (labeled as Si_1_) and one tetracoordinate silicon atom (labeled as Si_2_). Additionally, there are four saturated oxygen atoms (labeled as O_1_) and two unsaturated oxygen atoms (labeled as O_2_) in this cluster. Among them, Si_1_ and O_2_ atoms are located at the termination of the cluster, whereas Si_2_ and O_1_ atoms are positioned in the central region. Figure 1b illustrates the structure of the (SiO_2_)_3_O_2_H_4_ cluster, which is characterized by the existence of four hydroxyl groups connecting with terminal Si_1_ atoms. Different from the (SiO_2_)_3_ cluster, the (SiO_2_)_3_O_2_H_4_ cluster is composed of three tetracoordinate Si_1_ and Si_2_ atoms along with six fully saturated O_1_ and O_2_ atoms. 

### 2.2. The Formation of 2-Chlorothiophenolate on Silica Clusters

Figure 2a illustrates the formation of 2-chlorothiophenolate from the initial 2-CTP on the (SiO_2_)_3_ cluster embedded with potential barriers Δ*E* (in kcal/mol) and reaction heats Δ*H* (in kcal/mol). Figure 2b supplements the corresponding potential energy profile. The formation of 2-chlorothiophenolate consists of two elementary steps, the adsorption of 2-CTP on the (SiO_2_)_3_ cluster and the dissociation of H atom of the sulfhydryl group in the adsorbed 2-CTP (S-H dissociation), finally leading the formation of 2-chlorothiophenolate. This can be supported by experimental study, in which the generation of 2,3,6-trichlorophenoxy radicals via hydrogen abstraction from 2,3,6-trichlorophenols was elucidated during the PCDD formation process over a Cu(II)O/silica matrix [20]. As shown in Figure 2a, three possible pathways (pathways 1−3) were considered for the adsorption and S-H dissociation of 2-CTP on the (SiO_2_)_3_ cluster. For the adsorption process, three adsorption complexes, IM1, IM3 and IM5, were obtained from pathway 1, pathway 2 and pathway 3 with adsorption energies of −22.74 kcal/mol, −2.42 kcal/mol and −22.87 kcal/mol, respectively. The 2-CTPs in IM1 and IM5 adsorb on the Si_1_ atoms of (SiO_2_)_3_ cluster (S-Si_1_ coupling), while the 2-CTP in IM3 attaches on the Si_2_ atom of (SiO_2_)_3_ cluster (S-Si_2_ coupling). For the subsequent S-H dissociation (Figure 2a), in pathway 1 and pathway 2, the H atoms in both IM1 and IM3 transfer from the S atom of the initial 2-CTP to the O_1_ atoms of the (SiO_2_)_3_ cluster. Simultaneously, the O_1_-Si_1_ bond in IM1 and the O_1_-Si_2_ bond in IM3 are broken. This simultaneous hydrogen transfer and bond-breaking process lead to the formation of IM2 in pathway 1 and IM4 in pathway 2. Differently, in pathway 3, the S-H bond of IM5 is broken with the H atom directly attached to the O_2_ atom, leading to the formation of IM6.

We further explored the formation of 2-chlorothiophenolate from the initial 2-CTP on the hydroxylated silica cluster ((SiO_2_)_3_O_2_H_4_ cluster), as illustrated in Figure 3a,b. Similar to the (SiO_2_)_3_ cluster, the adsorption and S-H dissociation of 2-CTP to form 2-chlorothiophenolate on the (SiO_2_)_3_O_2_H_4_ cluster can take place through pathways 4−6. Pathway 4 and pathway 5 are similar to pathway 1 and pathway 2, respectively, leading to the formation of IM8 and IM10. However, pathway 6 is slightly different from pathway 3. Specifically, after the 2-CTP adsorbs on the (SiO_2_)_3_O_2_H_4_ cluster via S-Si_1_ coupling to form IM11, the H atom in IM11 is detached and connects with the hydroxyl group of the (SiO_2_)_3_ cluster to eliminate H_2_O, resulting in the formation of IM12. This formation mechanism is in concordance with the experimental observation from an in situ FTIR measurement, in which the formation of 2-chlorophenolate on silica is related to the loss of surface hydroxyl groups [36]. Additionally, Figure 4 presents all optimized geometries of intermediates and transition states involved in the formation of 2-chlorothiophenolate from initial 2-CTP on the (SiO_2_)_3_ cluster and (SiO_2_)_3_O_2_H_4_ cluster, and Appendix A lists crucial parameters for transition states.

### 2.3. Formation of Pre-PCDT Intermediates via the L–H Mechanism

Following the formation of 2-chlorothiophenolate in IM6 via the most favorable pathway 3, the (SiO_2_)_3_ cluster can continue to adsorb the second 2-CTP to form another 2-chlorothiophenolate. Then, the coupling of two formed 2-chlorothiophenolates leads to the formation of pre-PCDT intermediates via the L–H mechanism. Figure 5 and Figure 6 demonstrate the formation routes of pre-PCDT intermediates embedded with potential barriers Δ*E* (kcal/mol) and reaction heats Δ*H* (kcal/mol) from the coupling of two adsorbed 2-chlorothiophenolates via the L–H mechanism on the (SiO_2_)_3_ cluster. As shown in Figure 5, the second 2-CTP adsorbs on the Si_2_ atom in IM6 with an adsorption energy of 9.17 kcal/mol, resulting in the formation of IM13. Subsequently, the second 2-CTP adsorbed in IM13 undergoes two kinds of S-H dissociation to form the second 2-chlorothiophenolate. In pathway 7, the detached H atom transfers from the S atom of the second 2-CTP to the O_1_ atom of the (SiO_2_)_3_ cluster, where O_1_ is far away from the initial 2-CTP and the Si_2_-O_1_ bond is broken at the same time, resulting in the formation of IM14. In pathway 8, the H atom detaches from the S atom of the second 2-CTP to the O_1_ atom of the (SiO_2_)_3_ cluster, where O_1_ is located between the two formed 2-chlorothiophenolates, and the Si_2_-O_1_ bond is broken immediately, resulting in the formation of IM15.

The subsequent reactions of IM14 shown in Figure 6a primarily proceed via the *ortho–ortho* coupling of 2-chlorothiophenolates to form pre-PCDT intermediates. Three C/C coupling modes are proposed based on three possible pathways (pathways 9−11). In the initial step of pathway 9, a simultaneous process occurs involving the coupling of two carbon (hydrogen)-centered radical mesomeres (CH/CH for short) and the abstraction of a hydrogen atom connected with the ortho carbon by another H atom (H abstraction). This concurrent reaction results in the formation of the pre-PCDT intermediate, IM16. In the subsequent step, the IM16 undergoes a five-membered ring H shift, where the H atom originally attached with the ortho carbon atom of the initial 2-CTP moves to the S atom of the second 2-CTP. Meanwhile, the S-C bond is broken, leading to the formation of the pre-PCDT intermediate, IM17. In pathway 10, the carbon (hydrogen)-centered radical mesomere recombines with the carbon (chlorine)-centered radical mesomere (CH/CCl for short) and the Cl atom connected with ortho C is eliminated simultaneously (Cl elimination), forming the pre-PCDT intermediate, IM18. In pathway 11, the coupling of two carbon (chlorine)-centered radical mesomeres (CCl/CCl for short) and the Cl elimination simultaneously take place to generate the pre-PCDT intermediate, IM19.

Similar to the subsequent reactions of IM14, as shown in Figure 6b, IM15 can undergo CH/CCl and CCl/CCl couplings, accompanied by the Cl eliminations, finally resulting in the formation of IM23 in pathway 13 and IM24 in pathway 14, respectively. Differently, in pathway 12, the H abstraction step and C/CH coupling of IM15 need to occur separately, which is not a synergistic reaction. Thus, pathway 12 has one more step than pathway 9. Finally, pathway 12 concludes with the H shift step, forming the pre-PCDT intermediate, IM22.

### 2.4. Formation of Pre-PCTA/DT Intermediates via the E–R Mechanism

In addition to the L–H mechanism, the initially formed 2-chlorothiophenolate can directly react with the gas-phase 2-CTP to generate pre-PCTA/DT intermediates via the E–R mechanism. Figure 7 demonstrates the formation routes of pre-PCTA/DT intermediates embedded with the potential barriers Δ*E* (kcal/mol) and reaction heats Δ*H* (kcal/mol) from the coupling of adsorbed 2-chlorothiophenolate on (SiO_2_)_3_ cluster with gas-phase 2-CTP via the E–R mechanism. In Figure 7, pathway 15 and pathway 16 are similar, in which C/C couplings occur to form pre-PCDT intermediates. Specifically, in pathway 15, CH/CCl coupling and the Cl elimination occur synergistically, resulting in the formation of the pre-PCDT intermediate, IM25. In pathway 16, CCl/CCl coupling and the Cl elimination occur simultaneously, resulting in the formation of a pre-PCDT intermediate, IM26. Pathway 17 and pathway 18 are identical, in which S/C couplings take place to form pre-PCTA intermediates. In pathway 17, the recombination of the sulfur-centered mesomere with the carbon (hydrogen)-centered radical mesomere (S/CH for short) and H_2_ elimination take place simultaneously to form the pre-PCTA intermediate, IM27. In pathway 18, the coupling of the sulfur-centered mesomere with the carbon (chlorine)-centered radical mesomere (S/CCl for short) and Cl elimination occur simultaneously to form the pre-PCTA intermediate, IM28. Appendix A shows all the optimized structures of important transition states in the L–H and E–R mechanisms. In order to compare the formation mechanism of pre-PCTA/DT intermediates on (SiO_2_)_3_ with that on (SiO_2_)_3_O_2_H_4_ clusters, the formation routes of pre-PCTA/DT intermediates from the coupling of adsorbed 2-chlorothiophenolate on the (SiO_2_)_3_O_2_H_4_ cluster with gas-phase 2-CTP via the E–R mechanism are listed in Appendix A of the Appendix A. Similarly, four pre-PCTA/DT intermediates (IM34−IM37) can be obtained from four pathways (pathways 23−26). In order to investigate the effect of entropy on the formation of pre-PCTA/DT intermediates, Appendix A illustrates the Gibbs free energy Δ*G* (kcal/mol) and free energy barrier Δ*G^≠^* (kcal/mol) for the crucial elementary reactions involved in the formation of pre-PCTA/DT intermediates from 2-CTP on (SiO_2_)_3_ and (SiO_2_)_3_O_2_H_4_ clusters.

### 2.5. Rate Constant Calculations

Due to the limitation of the detection methods, it is challenging to measure the rate constants of elementary reactions in experimental research. Theoretical methods can provide powerful support for accurate calculations of rate constants. This study calculated the rate constants for the main elementary steps involved in the heterogeneous formation of pre-PCTA/DT intermediates from 2-CTP on the (SiO_2_)_3_ cluster and (SiO_2_)_3_O_2_H_4_ clusters, with the temperature ranging from 600 K to 1200 K. The corresponding values are calculated and listed in Appendix A. The relationship between the reaction rate constant and temperature was molded using the Arrhenius formulas, as illustrated in Table 1, providing the relevant factors, pre-exponential factors and activation energies.

## 3. Discussion

### 3.1. The Formation of 2-Chlorothiophenolate on Silica Clusters

Three pathways have been designed to explore the formation of 2-chlorothiophenolate from the initial 2-CTP on the (SiO_2_)_3_ cluster, as shown in Figure 2. The IM5 (adsorption energy −22.87 kcal/mol) has much lower adsorption energy than IM1 (adsorption energy −22.74 kcal/mol) and IM3 (adsorption energy −2.42 kcal/mol). Consequently, IM5 in pathway 3 demonstrates a greater formation potential and exhibits more stability than IM1 in pathway 1 and IM3 in pathway 2. Additionally, in subsequent S-H dissociation steps, pathway 3 requires crossing a lower potential barrier of 7.58 kcal/mol than the values of 22.01 kcal/mol and 13.33 kcal/mol in pathway 1 and pathway 2, respectively. Moreover, pathway 3 (reaction heat −43.97 kcal/mol) exhibits higher exothermicity than pathway 1 (reaction heat −2.01 kcal/mol) and pathway 2 (reaction heat −24.70 kcal/mol). Therefore, pathway 3 is more thermodynamically favorable than pathway 1 and pathway 2, and the formation of IM6 is preferred over IM2 and IM4. These imply that the terminal Si_1_ atom of the (SiO_2_)_3_ cluster is more prone to the adsorption of 2-CTP compared to the middle Si_2_ atom, and the O_2_ atom is more conducive to the S-H dissociation of adsorbed 2-CTP to form 2-chlorothiophenolate than the O_1_ atom. These findings are consistent with results obtained in a prior theoretical investigation conducted by Pan et al., which focused on the reaction of 2-CP catalyzed by the (SiO_2_)_3_ cluster [37]. It is worth noting that a previous study by Xu et al. reported that the S-H dissociation of 2-CTP in the gas phase is a highly endothermic process, with a value of 86.51 kcal/mol [41]. However, the dissociation process undergoes a significant shift towards exothermicity in the presence of a (SiO_2_)_3_ cluster. Consequently, the existence of the (SiO_2_)_3_ cluster can facilitate the dissociation of the S-H bond of the adsorbed 2-CTP to form 2-chlorothiophenolate to some extent. For the formation of 2-chlorothiophenolates from 2-CTPs on (SiO_2_)_3_O_2_H_4_, pathways 4−6 are identified, as shown in Figure 3. Based on a similar analysis with the (SiO_2_)_3_ cluster, it can be concluded that the adsorption and S-H dissociation of 2-CTP on the (SiO_2_)_3_O_2_H_4_ cluster preferentially proceed through pathway 5 instead of pathway 4 and pathway 6, resulting in a preferred formation of IM10 compared to IM8 and IM12. It can be inferred that the 2-CTP is more likely to adsorb on the middle Si_2_ atom than the terminal Si_1_ atom in the (SiO_2_)_3_O_2_H_4_ cluster, and the O_1_ atom exhibits higher activity for the formation of 2-chlorothiophenolate compared to the O_2_ atom. These are different from the (SiO_2_)_3_ cluster discussed above. However, similar to the (SiO_2_)_3_ cluster, the presence of the (SiO_2_)_3_O_2_H_4_ cluster also exhibits mild catalysis for the conversion of 2-CTP to 2-chlorothiophenolate. 

Upon comparing pathway 3 with pathway 5, it is evident that the adsorption energy for IM5 in pathway 3 is −22.87 kcal/mol, which is notably lower than the value of −3.33 kcal/mol for IM9 involved in pathway 5. In addition, the subsequent S-H dissociation in pathway 3 carries a lower potential barrier than that of pathway 5 by 17.96 kcal/mol. Obviously, pathway 3 is favored over pathway 5, suggesting that the (SiO_2_)_3_ cluster is more effective in catalyzing the formation of 2-chlorothiophenolate than the (SiO_2_)_3_O_2_H_4_ cluster. This may be attributed to the presence of unsaturated O_2_ and Si_1_ atoms on the terminal sites of the (SiO_2_)_3_ cluster, which enhances the reactivity of the (SiO_2_)_3_ cluster. However, the hydroxyl groups in the (SiO_2_)_3_O_2_H_4_ cluster are saturated, resulting in a relative inertness towards the reaction with 2-CTP. However, (SiO_2_)_3_O_2_H_4_ clusters, under high-temperature conditions, can transform into (SiO_2_)_3_ clusters, which exhibit higher catalytic activity and promote the conversion of 2-CTP to 2-chlorothiophenolate.

### 3.2. Formation of Pre-PCDT Intermediates via the L–H Mechanism

After the formation of 2-chlorothiophenolate in IM6 via the most favorable pathway 3, the second 2-CTP can adsorb on the (SiO_2_)_3_ cluster via S-Si_2_ coupling to form the second 2-chlorothiophenolat (Figure 5). A comparison of pathway 7 with pathway 8 shows that the S-H dissociation step in pathway 7 has a slightly lower potential barrier of 10.75 kcal/mol than the value of 11.13 kcal/mol in the S-H dissociation step in pathway 8. In addition, the S-H dissociation step in pathway 7 is more exothermic (−27.34 kcal/mol) than the S-H dissociation step in pathway 8 (−23.07 kcal/mol). So, pathway 7 is mildly favored over pathway 8. The formation potential of IM14 is slightly higher than that of IM15. 

Figure 6a shows the pre-PCDT formation from the coupling of two 2-chlorothiophenolates via the L–H mechanism. The ranking for the potential barrier values of the three C-C coupling modes is as follows: CH/CH coupling and H abstraction (66.74 kcal/mol) > CH/CCl coupling and Cl elimination (85.20 kcal/mol) > CCl/CCl coupling and Cl elimination (106.04 kcal/mol). In addition, the CH/CH coupling and H abstraction step in pathway 9 are exothermic by -8.82 kcal/mol, while the CH/CCl coupling and Cl elimination in pathway 10 and CCl/CCl coupling and Cl elimination in pathway 11 are endothermic by 76.69 kcal/mol and 91.62 kcal/mol, respectively. Thus, pathway 9 is energetically more favorable than pathways 10 and 11 in Figure 6a, resulting in the formation of IM17. For the same reason, pathway 12 is more energetically favorable than pathways 13 and 14 in Figure 6b, resulting in the formation of IM22. 

In pathway 9, the CH/CH coupling and H abstraction step have the highest potential barrier of 66.74 kcal/mol and represent the rate-determining step. Similarly, the H abstraction step is the rate-determining step of pathway 12. Although pathway 9 has one step less than pathway 12, the rate-determining step of pathway 9 presents a much higher potential barrier than that in pathway 12. Thus, pathway 9 and pathway 12 are competitive. Both IM17 and IM22 are possible pre-PCDT intermediates from the coupling of two adsorbed 2-chlorothiophenolates on the (SiO_2_)_3_ cluster via the L–H mechanism. The formed pre-PCDTs can subsequently undergo the ring-close reactions, finally leading to the formation of PCDTs. These theoretical results can provide reasonable explanations for the experimental results in which the L–H mechanism mainly contributes to the formation of PCDTs.

### 3.3. Formation of Pre-PCTA/DT Intermediates via the E–R Mechanism

For the pre-PCDT formation via the E–R mechanism in Figure 7, the CH-CCl coupling and Cl elimination step in pathway 15 have a potential barrier of 71.11 kcal/mol, which is lower than the value of 78.97 kcal/mol of the CCl-CCl coupling and Cl elimination step in pathway 16. In addition, the CH-CCl coupling and Cl elimination step in pathway 15 (70.04 kcal/mol) are less endothermic than the CCl-CCl coupling and Cl elimination step in pathway 16 (73.33 kcal/mol). Thus, pathway 15 is favored over pathway 16. For the same reason, for the pre-PCTA formation via the E–R mechanism in Figure 7, pathway 18 is favored over pathway 17. A comparison of pathway 15 with pathway 18 shows that the potential barrier in pathway 18 is much lower than that in pathway 15 by about 17 kcal/mol. Thus, pathway 18 is favored over pathway 15. Meanwhile, the formation potential of IM28 (pre-PCTA intermediate) is higher than that of IM25 (pre-PCDT intermediate). In other words, the pre-PCTA formation is much easier than the pre-PCDT formation from the 2-CTP as a precursor on the (SiO_2_)_3_ cluster via the E–R mechanism. The formed pre-PCTAs can undergo HCl elimination to form the ring-structural PCTAs, which agrees well with the experimental observations where the E–R mechanism is responsible for the formation of PCTAs. A similar conclusion can be reached for the pre-PCTA/DT formation on the (SiO_2_)_3_O_2_H_4_ clusters via the E–R mechanism in Appendix A. 

A comparison of Figure 5, Figure 6 and Figure 7 shows that the rate-determining step involved in the formation of pre-PCDTs via the L–H mechanism requires a similar potential barrier as that involved in the formation of pre-PCTAs via the E–R mechanism. However, the preferred formation routes of pre-PCDTs via the L–H mechanism involve four or five elementary steps, whereas the formation of pre-PCTAs via the E–R mechanism involves one elementary process. Thus, the formation of pre-PCTAs via the E–R mechanism is preferred over the formation of pre-PCDTs from the L–H mechanism. This conclusion may be supported indirectly by the experimental results: PCDDs are the major dioxin products, rather than PCDFs, in the heterogeneous pyrolysis of 2-CP by neat silica [35]. 

In general, the formation of pre-PCTA/DTs from 2-CTP needs to occur cross a 50–60 kcal/mol potential barrier with a catalyst of silica clusters. This implies that while the silica can catalyze the formation of PCTA/DTs from 2-CTP, its catalytic activity is relatively mild. Considering the high concentration of silica in fly ash, the catalyst effect of silica should not be ignored. 

### 3.4. Rate Constant Calculations

The results obtained from the rate constant are consistent with the thermodynamic analysis discussed above. For instance, as shown in Appendix A, at 800 K, the rate constants for the S-H dissociation in pathways 1−3 are 7.62 × 10^6^ s^−1^ (TS1), 1.99 × 10^8^ s^−1^ (TS2) and 3.28 × 10^11^ s^−1^ (TS3), respectively. The highest rate constant value for pathway 3 reveals its preference over pathway 1 and pathway 2. This reaffirms the energetic conclusion that pathway 3 is favored over pathway 1 and pathway 2. Similarly, at 1200 K, as shown in S2, the rate constant for the S-H dissociation in pathway 5 is 8.99 × 10^5^ s^−1^ (TS5), which is larger than the values of 8.36 × 10^6^ s^−1^ (TS4) in pathway 4 and 1.52 × 10^5^ s^−1^ (TS6) in pathway 6. This is in accordance with the thermal analysis in which pathway 5 is preferred over pathway 4 and pathway 6. 

The consistency between kinetic and energetic analysis can also be reached for the subsequent formation of pre-PCTA/DT intermediates through the L–H and E–R mechanisms. For a given temperature, the rate constant of Cl elimination, and the S-C coupling step involved in pathway 18, is larger than that of the corresponding steps involved in pathways 15−17. For example, at 1000 K, the rate constant for the Cl elimination and S-C coupling step in pathway 18 is 4.96 × 10^−27^ s^−1^ (TS21), which is 4−11 orders of magnitude larger than the values involved in pathways 15−17. It can be deduced that the E–R mechanism accounts for the formation of pre-PCTAs, which is in line with the thermodynamic analysis. The same conclusions can be reached for the formation of pre-PCDT intermediates through the L–H mechanism. In general, these consistencies can be reached for all crucial elementary reactions involved in the heterogeneous reaction of PCTA/DTs from 2-CTP on silica clusters.

## 4. Materials and Methods

### 4.1. Density Functional Theory

All high-accuracy quantum calculations were carried out using the Gaussian 09 software package (Wallingford, CT, USA) [42]. The MPWB1K method, which is based on the modified Perdew and Wang exchange functional (MPW) and Becke’s correlation functional (B95), has been used for the prediction of transition geometry and thermochemical kinetics [43]. The geometrical parameters of reactants, intermediates, transition states and products involved in the formation of pre-PCTA/DTs from 2-CTP on (SiO_2_)_3_ and (SiO_2_)_3_O_2_H_4_ clusters were obtained at the MPWB1K level with a standard 6-31+G(d,p) basis set [43]. To determine the thermal contributions to the free energy, the harmonic vibrational frequencies of reactants, intermediates, transition states and products were also calculated at the same level to obtain the zero-point energy (ZPE). The harmonic vibrational frequency calculations show that the transition states have exactly one imaginary frequency, while the reactants, intermediates and products have no imaginary frequencies. Intrinsic reaction coordination (IRC) analysis was performed to obtain the minimum energy path (MEP) and verify that each transition state connects the designed reactants and products [44]. The single-point energies of reactants, intermediates, transition states and products were calculated using a more flexible basis set, 6-311+G(3df,2p), to yield more reliable reaction heats and potential barriers. The zero-point energy (ZPE) correction was finally included in the values of reaction heats and potential barriers.

### 4.2. Kinetic Calculations

Based on the TST theory with Wigner tunneling correction within the KiSThelP program [45], the rate constants of the crucial elementary steps involved in the formation of PCTA/DTs from 2-CTP on (SiO_2_)_3_ and (SiO_2_)_3_O_2_H_4_ clusters were calculated over the temperature range of 600−1200 K on the MPWB1K/6-31+G(d,p)//MPWB1K/6-311+G(3df,2p) level. In KiSThelP, the equation usually presented for the conventional TST is
(1)$$k^{TST}=\sigma \frac{k_{b}T}{h}\frac{Q^{TS}(T)}{N_{A}Q^{R}(T)}e^{\frac{V^{\neq }}{k_{b}T}}$$
where *σ* is the reaction path degeneracy, *k_b_* is Boltzmann’s constant, *T* is the temperature, *h* is Planck’s constant and *N_A_* is Avogadro’s number (it disappears for unimolecular reactions rate constants with units of s-1). *V^≠^* is the difference in zero-point excluded potential energy between the transition state (TS, assumed to be located at the saddle point on the PES) and the reactant(s) (zero-point energy contributions are included in the partition functions). *Q^TS^* and *Q^R^* denote the total partition functions of the *TS* and the reactant(s) with the translational partition functions expressed per unit volume. *Q^TS^* excludes the reaction coordinate.

### 4.3. Accuracy Verification

It is crucial to validate theoretical calculations for ongoing work. Initially, optimized geometries and vibrational frequencies of thiophenol, 4-chlorotriophenol and dibenzothiophene at the MPWB1K/6-31+G(d,p) level were compared with experimental data, showing consistent results within 1.0% for geometries and 9.0% for vibrational frequencies [46,47,48,49]. By employing the MPWB1K/6-311+G(3df,2p)//MPWB1K/6-31+G(d,p) level, reaction enthalpy for thiophenol + thiophenol → dibenzothiophene + H_2_S + H_2_ was calculated, yielding *−*7.60 kcal/mol at 298.15 K and 1.0 atm, which is close to the experimental value (*−*7.74 kcal/mol) based on the measured standard enthalpies [50]. For the formation of 2-chlorothiophenolate from initial 2-CTP on the (SiO_2_)_3_ cluster (IM5 → IM6 via TS3), single-point energies for IM5, IM6 and TS3 were calculated at the B2PLYP/def2tzvp level. Moreover, the potential barrier and reaction heat for IM5 → IM6 via TS3 were calculated at the M062X/6*-*311*+G*(3df,2p)//M062X/6*-*31*+G*(d,p) level. All results align with those obtained by MPWB1K/6*-*311*+G*(3df,2p)//MPWB1K/6*-*31*+G*(d,p) with the relative deviation within 7%. Thus, the MPWB1K/6*-*311*+G*(3df,2p)//MPWB1K/6*-*31*+G*(d,p) level ensures accurate calculations for heterogenous PCTA/DT formation from 2-CTP precursors on silica clusters.

## 5. Conclusions

In this work, we have investigated the heterogeneous formation mechanism of pre-PCTA/DTs from 2-CTP on the dehydrated and hydroxylated silica clusters through DFT calculations. Moreover, the rate constants for the main elementary steps involved in the heterogonous formation of pre-PCTA/DTs were calculated over the temperature range of 600–1200 K. The following conclusions were drawn:
(1)The adsorption and S-H dissociation of 2-CTP to form 2-chlorothiophenolate preferentially occur at the end of the dehydrated silica cluster, while those processes primarily take place in the middle of the hydroxylated silica cluster. Comparatively, the dehydrated silica cluster is more effective in catalyzing the conversion of 2-CTP to 2-chlorothiophenolate than the hydroxylated silica cluster.(2)2-CTP, as a precursor adsorbed on the dehydrated silica cluster, can lead to the formation of pre-PCDTs via the L–H mechanism, while the E–R mechanism can result in the generation of pre-PCTAs. The formation of pre-PCTAs via the E–R mechanism is more prone to occur than the formation of pre-PCDTs through the L–H mechanism. (3)The silica is a relatively mild catalyst that can facilitate the conversion of 2-CTP to pre-PCTA/DTs. However, considering the high concentration of silica in fly ash, the catalytic effect of silica cannot be ignored.

## Figures and Tables

**Figure 1 ijms-25-03485-f001:**
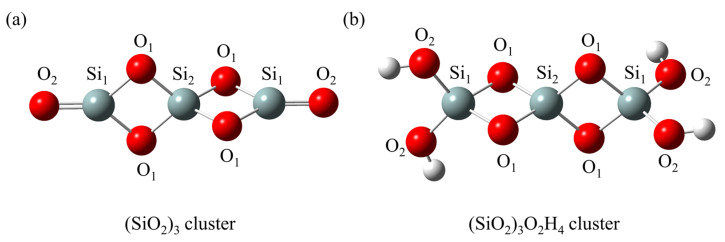
Optimized geometries for (**a**) the dehydrated silica cluster ((SiO_2_)_3_) and (**b**) the hydroxylated silica cluster ((SiO_2_)_3_O_2_H_4_) at the MPWB1K/6-31+G(d,p) level. Red, gray and white spheres represent O, Si and H atoms, respectively.

**Figure 2 ijms-25-03485-f002:**
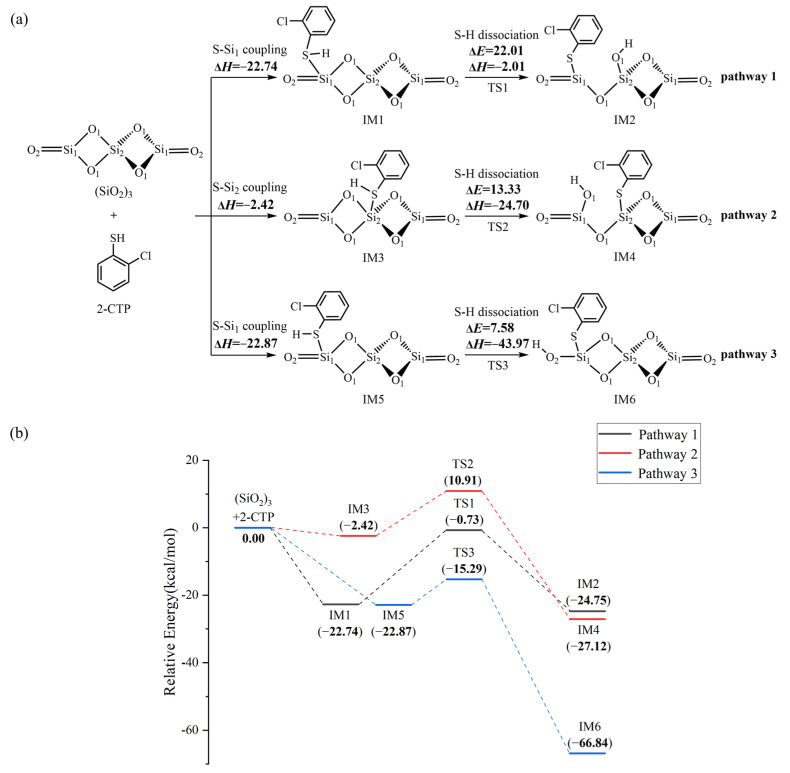
(**a**) The formation of 2-chlorothiophenolate from initial 2-CTP on the (SiO_2_)_3_ cluster embedded with the potential barriers Δ*E* (in kcal/mol) and reaction heats Δ*H* (in kcal/mol); (**b**) potential energy profile for the formation of 2-chlorothiophenolate from initial 2-CTP on the (SiO_2_)_3_ cluster.

**Figure 3 ijms-25-03485-f003:**
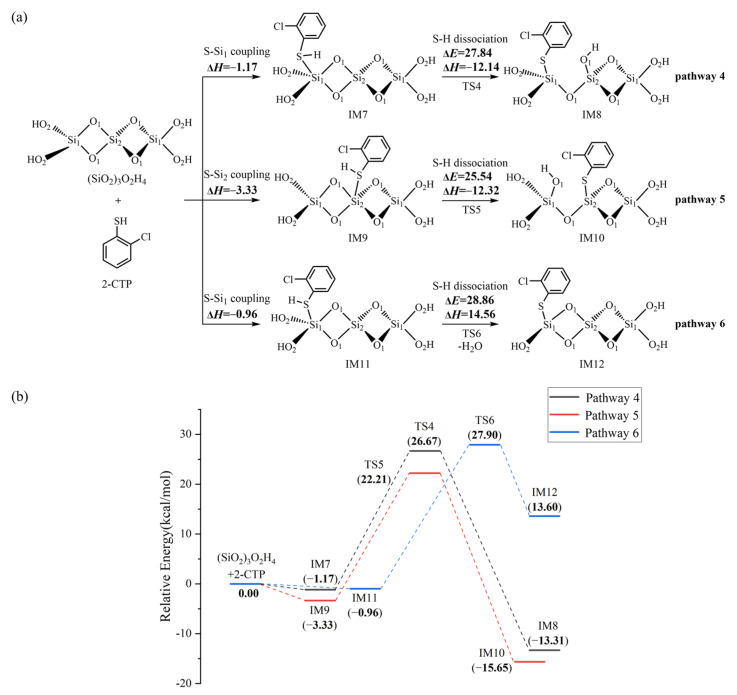
(**a**) The formation of 2-chlorothiophenolate from initial 2-CTP on the (SiO_2_)_3_O_2_H_4_ cluster embedded with the potential barriers Δ*E* (in kcal/mol) and reaction heats Δ*H* (in kcal/mol); (**b**) the potential energy profile for the formation of 2-chlorothiophenolate from initial 2-CTP on the (SiO_2_)_3_O_2_H_4_ cluster.

**Figure 4 ijms-25-03485-f004:**
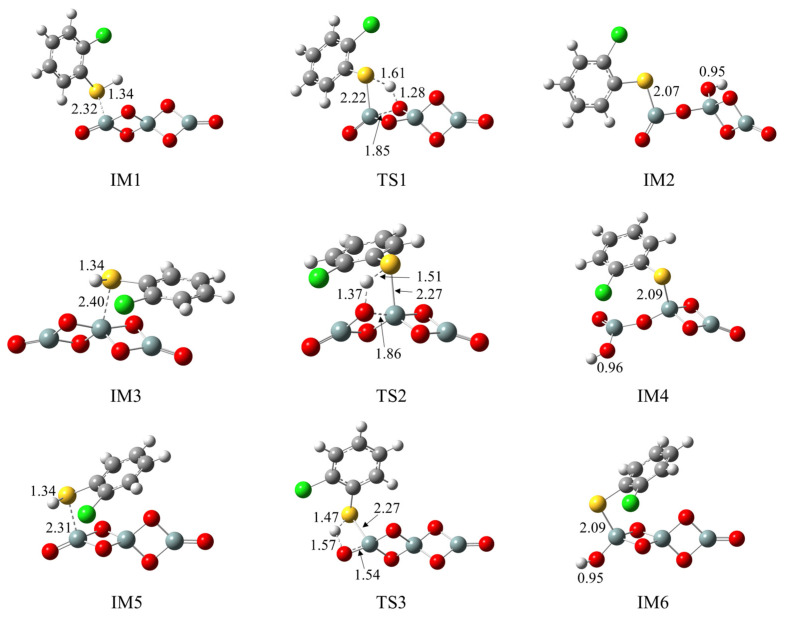
Optimized geometries involved in the formation of 2-chlorothiophenolate from initial 2-CTP on the (SiO_2_)_3_ cluster and the (SiO_2_)_3_O_2_H_4_ cluster.

**Figure 5 ijms-25-03485-f005:**
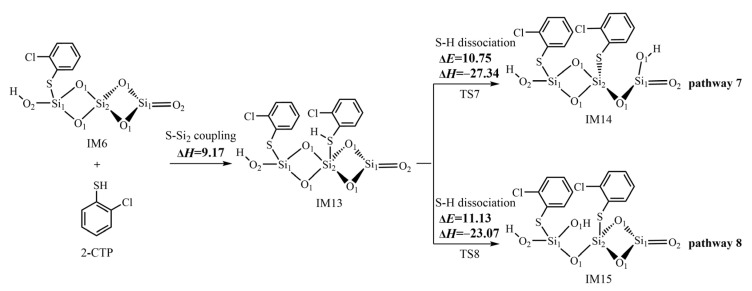
The formation of the second 2-chlorothiophenolate from the second 2-CTP on the (SiO_2_)_3_ cluster embedded with potential barriers Δ*E* (kcal/mol) and reaction heats Δ*H* (kcal/mol).

**Figure 6 ijms-25-03485-f006:**
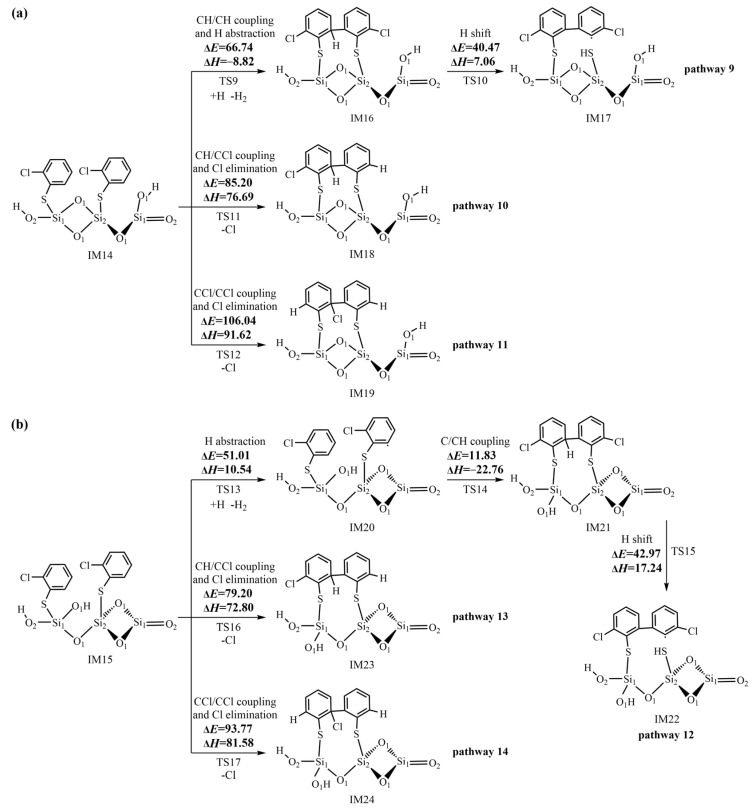
Formation routes of pre-PCDT intermediates embedded with potential barriers Δ*E* (kcal/mol) and reaction heats Δ*H* (kcal/mol) from the coupling of two adsorbed 2-chlorothiophenolates on the (SiO_2_)_3_ cluster via the L–H mechanism starting with (**a**) IM 14 and (**b**) IM15.

**Figure 7 ijms-25-03485-f007:**
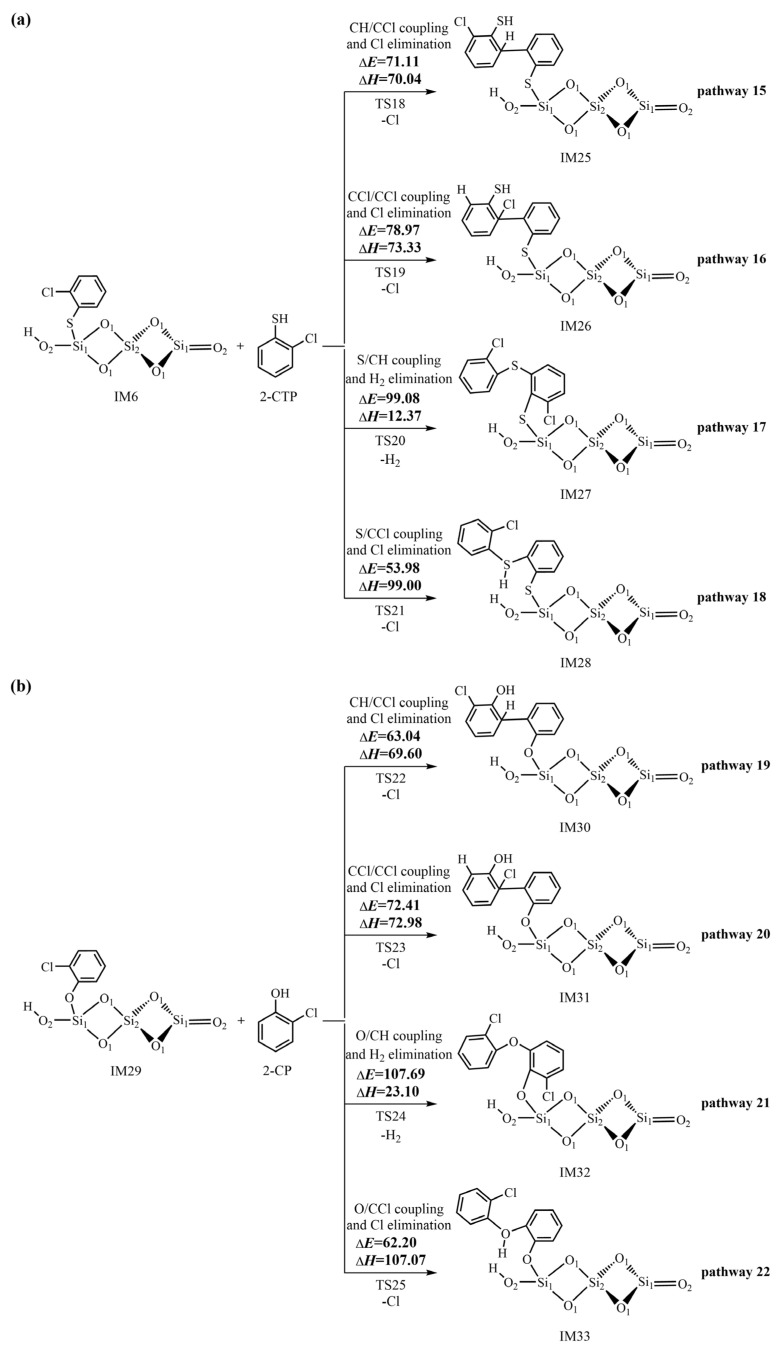
Formation routes of pre-PCTA/DT intermediates embedded with the potential barriers Δ*E* (kcal/mol) and reaction heats Δ*H* (kcal/mol) from the coupling of adsorbed 2-chlorothiophenolate on the (SiO_2_)_3_ cluster with gas-phase 2-CTP via the E–R mechanism starting with (**a**) IM16 and (**b**) IM29.

**Table 1 ijms-25-03485-t001:** Arrhenius equations for key elementary reactions involved in the formation of pre-PCTA/DT intermediates from 2-CTP on (SiO_2_)_3_ and (SiO_2_)_3_O_2_H_4_ clusters. (Units are s^−1^ and cm^−3^ molecule^−1^ s^−1^ for unimolecular and bimolecular reactions, respectively).

Reactions	Arrhenius Formulas
IM1 → IM2 via TS1	*k*(T) = (8.63 × 10^12^) exp (−11,138/T)
IM3 → IM4 via TS2	*k*(T) = (8.24 × 10^11^) exp (−6651/T)
IM5 → IM6 via TS3	*k*(T) = (4.55 × 10^13^) exp (−3936/*T*)
IM7 → IM8 via TS4	*k*(T) = (8.84 × 10^11^) exp (−13,886/*T*)
IM9 → IM10 via TS5	*k*(T) = (3.33 × 10^10^) exp (−12,643/*T*)
IM11 → IM12 via TS6	*k*(T) = (1.63 × 10^10^) exp (−13,915/*T*)
IM13 → IM14 via TS7	*k*(T) = (1.46 × 10^13^) exp (−5576/*T*)
IM13 → IM15 via TS8	*k*(T) = (1.53 × 10^12^) exp (−5482/*T*)
IM14 → IM18 + Cl via TS11	*k*(T) = (1.61 × 10^12^) exp (−43,242/*T*)
IM14 → IM19+ Cl via TS12	*k*(T) = (3.10 × 10^12^) exp (−53,879/*T*)
IM15 → IM20 + H_2_ via TS13	*k*(T) = (5.44 × 10^−14^) exp (−25,324/*T*)
IM15 → IM23 + Cl via TS16	*k*(T) = (1.16 × 10^12^) exp (−40,134/*T*)
IM15 → IM24 + Cl via TS17	*k*(T) = (3.19 × 10^11^) exp (−47,477/T)
IM6 + 2-CTP → IM25 + Cl via TS18	*k*(T) = (1.03 × 10^−14^) exp (−37,954/*T*)
IM6 + 2-CTP → IM26 + Cl via TS19	*k*(T) = (5.02 × 10^−15^) exp (−41,857/*T*)
IM6 + 2-CTP → IM27 + H_2_ via TS20	*k*(T) = (1.88 × 10^−15^) exp (−52,003/*T*)
IM6 + 2-CTP → IM28 + Cl via TS21	*k*(T) = (2.75 × 10^−14^) exp (−29,308/*T*)
IM29 + 2-CP → IM30 + Cl via TS22	*k*(T) = (7.78 × 10^−16^) exp (−33,859/*T*)
IM29 + 2-CP → IM31 + Cl via TS23	*k*(T) = (7.83 × 10^−16^) exp (−38,866/*T*)
IM29 + 2-CP → IM32 + H_2_ via TS24	*k*(T) = (1.61 × 10^−16^) exp (−56,864/*T*)
IM29 + 2-CP → IM33 + Cl via TS25	*k*(T) = (2.05 × 10^−15^) exp (−33,514/*T*)

## Data Availability

Data will be made available on request.

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
