# Peer review of "Formation of Pre-PCTA/DT Intermediates from 2-Chlorothiophenol on Silica Clusters: A Quantum Mechanical Study"

_ijms, 2024, doi:10.3390/ijms25063485_

Round 1
Reviewer 1 Report
Comments and Suggestions for Authors
The theme is actual and the manuscript is well-presented. I have only two comments:
1) Some minor Grammatical errors and Typos can be found in the article. Please, read carefully one more and correct the whole text.
2) Have you any experimental data to support the suggested adsorption mechanism? I believe that some direct measurement data (e.g. gas adsorption analysis combined with infrared spectroscopy) are useful to prove the suggestions.
Comments on the Quality of English LanguageSome minor Grammatical errors and Typos can be found.
Author Response
Response to Reviewer 1 Comments
Dear Editor:
We gratefully appreciate for your enthusiastic feedback and the additional insightful comments from the reviewers. These comments are very valuable and helpful for revising and improving our manuscript. We have studied these comments carefully and have made revision which marked up using the “Track Changes” and “Highlight” function in the revised manuscript.
We greatly appreciate the chance to publish our manuscript on International Journal of Molecular Sciences. Thank you very much!
The comments of reviewer 1 and our replies are listed as follows:
Reviewer #1:
The theme is actual and the manuscript is well-presented. I have only two comments:
Comment 1. Some minor Grammatical errors and Typos can be found in the article. Please, read carefully one more and correct the whole text.
Response: Thank you for your valuable feedback on our manuscript. We are sorry for the grammar mistakes and have made further improvements to enhance the language quality of the manuscript. We have conducted additional polishing and refinement to ensure more accurate language and a foreign researcher also assisted in checking and polishing the manuscript again to avoid grammar mistakes and the quality of our language. Besides, we have also reviewed the use of professional terminology and abbreviation and refine expressions to ensure accuracy and consistency throughout the manuscript. Moreover, we have provided clearer and more precise descriptions of our computational methods and results to facilitate better understanding for readers. All revisions are made with the “track changes”.
Comment 2. Have you any experimental data to support the suggested adsorption mechanism? I believe that some direct measurement data (e.g. gas adsorption analysis combined with infrared spectroscopy) are useful to prove the suggestions.
Response: Thank you very much for this valuable comment. In response, we have searched some references related to this comment. Alderman et al. [36] observed the formation of 2-chlorophenolate by an in situ FTIR measurement during chemisorption of 2-chlorophenol on the surface of silica in a temperature range between 200 and 500 ℃. More interestingly, the FTIR difference spectrum obtained from the exposure of silica to 2-chlorophenol for 30 minutes at 325°C reveals a negative peak at 3747 cm-1, indicating a loss of surface hydroxyl groups on the silica. This experimental observation can prove our finding that after the 2-CTP adsorbs on the (SiO2)3O2H4 cluster via S-Si1 coupling, the H atom of the adsorbed 2-CTP is detached and connects with the hydroxyl group of (SiO2)3 cluster to eliminate H2O, resulting in the formation of 2-chlorothiophenolate. Besides, Yang et al [20] elucidated the generation of 2,3,6-trichlorophenoxy radicals via hydrogen abstraction from 2,3,6-trichlorophenol along the reaction pathways leading to PCDD formation over a Cu(II)O/silica matrix. This was verified by the appearance of g-signals in electron paramagnetic resonance (EPR) spectroscopy across a temperature range of 298−523 K. Moreover, Lomnicki et al. [25] have identified the presence of adsorbed chlorophenol radicals. Specifically, the electron paramagnetic resonance (EPR) results for copper oxide/silica before and after 2-MCP adsorption are depicted in Figure 6. In Figure 6a, the CuO/silica sample exhibits a broad signal originating from Cu(II) complexes. However, upon exposure to 2-MCP at 200°C for 2 minutes, a distinct sharp signal with a g-value of 2.0028 emerges in Figure 6b. This signal is attributed to the radical formed due to interactions between chlorophenol and the copper oxide surface. In summary, all the experimental results mentioned above can provide the background knowledge on the formation of 2-chlorothiophenolate on silica clusters from 2-chlorothiophenol in our study. The relevant explanations have been supplied in the manuscript.
Page 5, Line 161: “This can be supported by experimental study, in which the generation of 2,3,6-trichlorophenoxy radicals via hydrogen abstraction from 2,3,6-trichlorophenols were elucidated during PCDD formation process over a Cu(II)O/silica matrix [20].”
Page 7, Line 193: “This formation mechanism is in concordance with the experimental observation by an in situ FTIR measurement, in which the formation of 2-chlorophenolate on silica is related to the loss of surface hydroxyl groups [36].”
[36] Alderman, S. L.; Dellinger, B. FTIR investigation of 2-chlorophenol chemisorption on a silica surface from 200 to 500 C. J. Phys. Chem. A 2005, 109, 7725−7731.
[20] Yang, L.; Liu, G.; Zheng, M.; Zhao, Y.; Jin, R.; Wu, X.; Xu, Y. Molecular mechanism of dioxin formation from chlorophenol based on electron paramagnetic resonance spectroscopy. Environ. Sci. Tech. 2017, 51, 4999−5007.
[25] Lomnicki, S.; Dellinger, B. A detailed mechanism of the surface-mediated formation of PCDD/F from the oxidation of 2-chlorophenol on a CuO/silica surface. J. Phys. Chem. A 2003, 107, 4387−4395.

Reviewer 2 Report
Comments and Suggestions for Authors
In this paper, the authors revealed the detailed heterogeneous formation mechanism of pre-PCTA/DTs from 2-CTP on dehydrated and hydroxylated silica clusters through DFT calculations. The study is systematic and comprehensive with clear conclusions. This paper can be accepted in present form, however, there is one small error that could be corrected.
Line 211: "Three C/C coupling modes are proposed based on three possible pathways (pathways 9−10)." It should be pathways 9-11.
Author Response
Response to Reviewer 2 Comments
Dear Editor:
We gratefully appreciate for your enthusiastic feedback and the additional insightful comments from the reviewers. These comments are very valuable and helpful for revising and improving our manuscript. We have studied these comments carefully and have made revision which marked up using the “Track Changes” and “Highlight” function in the revised manuscript.
We greatly appreciate the chance to publish our manuscript on International Journal of Molecular Sciences. Thank you very much!
The comments of reviewer 2 and our replies are listed as follows:
Reviewer #2:
In this paper, the authors revealed the detailed heterogeneous formation mechanism of pre-PCTA/DTs from 2-CTP on dehydrated and hydroxylated silica clusters through DFT calculations. The study is systematic and comprehensive with clear conclusions. This paper can be accepted in present form, however, there is one small error that could be corrected.
Comment 1. Line 211: "Three C/C coupling modes are proposed based on three possible pathways (pathways 9−10)." It should be pathways 9-11.
Response: Thank you for your valuable feedback on our manuscript. We are sorry for the grammar mistakes and have made further improvements to enhance the language quality of the manuscript. We have conducted additional polishing and refinement to ensure more accurate language and a foreign researcher also assisted in checking and polishing the manuscript again to avoid grammar mistakes and the quality of our language. Besides, we have also reviewed the use of professional terminology and abbreviation and refine expressions to ensure accuracy and consistency throughout the manuscript. Moreover, we have provided clearer and more precise descriptions of our computational methods and results to facilitate better understanding for readers. All revisions are made with the “track changes”.
Page 7 Line 211: “Three C/C coupling modes are proposed based on three possible pathways (pathways 9−11).”

Reviewer 3 Report
Comments and Suggestions for Authors
Manuscript ID: ijms-2893407
Title: Formation of pre-PCTA/DT intermediates from 2-Chlorothiophenol on Silica Clusters: A Quantum Mechanical Study
Authors: Fei Xu *, Xiaotong Wang, Ying Li, Yongxia Hu, Ying Zhou, Mohammad Hassan Hadizadeh *
The reviewed manuscript investigates the adsorption processes of 2-chlorothiophenol on non-hydrated and hydrated silica clusters, including the following reaction between two adsorbed 2-chlorothiophenol molecules. The energy and structural parameters of these processes are determined, and the most probable reaction products are predicted. The study was conducted using standard DFT calculations. The general statement of the problem is reasonable and relevant.
However, the approach used by the authors has a critical flaw. The model silica clusters they employ are purely speculative. There is no evidence provided, not only that they can actually exist, but even that they can be considered as an acceptable approximation for describing a real silica surface. I see no argument that the results obtained may be useful or serve as a first step towards more thorough research. Thus, the conclusion made by the authors, stating that “silica is a relatively mild catalyst that can facilitate the conversion of 2-CTP to pre-PCTA/DTs. However, considering the high concentration of silica in fly ash, the catalytic effect of silica cannot be ignored” does not necessarily follow from the results obtained.
On the other hand, if the authors confine their examination solely to the proposed model clusters without making unfounded generalizations to real-life systems, then despite the internal limitations of the research and, consequently, the results obtained, I have no other comments on the article. Tables and figures are presented clearly, aiding in the comprehension of the text.
Author Response
Response to Reviewer 3 Comments
Dear Editor:
We gratefully appreciate for your enthusiastic feedback and the additional insightful comments from the reviewers. These comments are very valuable and helpful for revising and improving our manuscript. We have studied these comments carefully and have made revision which marked up using the “Track Changes” and “Highlight” function in the revised manuscript.
We greatly appreciate the chance to publish our manuscript on International Journal of Molecular Sciences. Thank you very much!
The comments of reviewer 3 and our replies are listed as follows:
Reviewer #3:
The reviewed manuscript investigates the adsorption processes of 2-chlorothiophenol on non-hydrated and hydrated silica clusters, including the following reaction between two adsorbed 2-chlorothiophenol molecules. The energy and structural parameters of these processes are determined, and the most probable reaction products are predicted. The study was conducted using standard DFT calculations. The general statement of the problem is reasonable and relevant.
However, the approach used by the authors has a critical flaw. The model silica clusters they employ are purely speculative. There is no evidence provided, not only that they can actually exist, but even that they can be considered as an acceptable approximation for describing a real silica surface. I see no argument that the results obtained may be useful or serve as a first step towards more thorough research. Thus, the conclusion made by the authors, stating that “silica is a relatively mild catalyst that can facilitate the conversion of 2-CTP to pre-PCTA/DTs. However, considering the high concentration of silica in fly ash, the catalytic effect of silica cannot be ignored” does not necessarily follow from the results obtained.
On the other hand, if the authors confine their examination solely to the proposed model clusters without making unfounded generalizations to real-life systems, then despite the internal limitations of the research and, consequently, the results obtained, I have no other comments on the article. Tables and figures are presented clearly, aiding in the comprehension of the text.
Response: We sincerely appreciate your invaluable feedback. Indeed, the silica cluster models we employed in this study are merely approximations of real surfaces and thus have inherent limitations. It is important to acknowledge that there are still challenges and constraints in terms of computational resources and methods for simulating the heterogeneous formation of PCDT/TAs catalyzed by surfaces, particularly when using software such as VASP. Given the substantial time, effort, and expertise required for simulating detailed heterogeneous formation of dioxin involving the complex surface, the use of silica cluster models can serve as a reasonable starting point. These models allow us to explore the intricate influence of neat silica on PCTA/DT formation, providing valuable insights despite their simplifications. There are some reasons why we chose the silica clusters models:
Firstly, the (SiO2)3 and (SiO2)3O2H4 clusters employed in this study contain two rhombic Si-O rings, which were confirmed to exist in the surface or the interior of amorphous and crystalline silica [39,40]. Besides, previous researches provide a background on our use of silica cluster in investigating the formation of dioxin-like compounds. Pan et al. [37] have corroborated the formation of adsorbed 2-chlorophenoxy radicals on neat and hydroxylated silica by employing the Si3O6 and Si3O8H4 clusters, respectively, via quantum chemical calculations. Mosallanejad et al. [35] studies partial oxidation of 2-chlorophenol on neat silica leading to catalytic formation of polychlorinated dibenzo-p-dioxins and polychlorinated dibenzofurans (PCDD/Fs) at temperatures of 250, 350, and 400 °C. Their study combined experimental simulation with computational methods. In their computational approach, silica clusters were utilized to elucidate the formation of pre-PCDD and pre-PCDF intermediates through the Eley-Rideal mechanism from the adsorption of 2-chlorophenol. These theoretical findings offer a plausible explanation and are highly consistent with the experimental results. Moreover, conclusions drawn from their application have shown agreement with some phenomena observed in experimental and theoretical researches. For example, this contribution concluded that the formation of pre-PCTAs via the E-R mechanism is more prone to occur than the formation of pre-PCDTs through the L-H mechanism, which is supported indirectly by the experimental results: PCDDs are the major dioxin products rather than PCDFs in the heterogeneous pyrolysis of 2-CP by neat silica. Our calculated results indicate that neat SiO2 serves as a mildly active catalyst for synthesizing PCDT/TAs from their precursors, which is in accordance with the experimental results [35]. These outcomes help elucidate certain aspects and offer valuable guidance for our research. Moving forward, we aim to use more accurate silica surfaces in the VASP package to investigate the heterogeneous formation mechanism of PCDT/TAs using first-principles calculations.
The relevant explanations have been refined in the manuscript:
Page 2, Line 94: “Mosallanejad et al. provided experimental and theoretical evidence that PCDD/Fs can be formed from a series of chlorophenols (CPs) with the mild catalysis of neat silica, even in the absence of transition metals [35].”
Page 2, Line 99: “Pan et al. [37] corroborated the formation of chemisorbed 2-chlorophenolate on dehydrated and hydroxylated silica clusters from a theoretical aspect.”
Page 3, Line 102: “It has been proved that employing the silica cluster model as the starting point is reasonable for investigating the complex influence of neat silica on PCTA/DT formation.”
Page 3, Line 129: “Figure 1 displays two types of silica clusters are employed: the dehydrated silica cluster ((SiO2)3) and the hydroxylated silica cluster ((SiO2)3O2H4), which have been proved reasonable for investigating the formation of dioxin-like compounds [35,37]. It is also important to note that the silica cluster models employed in this work are approximations of real surfaces. These models allow us to explore the intricate influence of neat silica on PCTA/DT formation, providing valuable insights despite their simplifications. It can be seen that both the (SiO2)3 and (SiO2)3O2H4 clusters contain two rhombic Si-O rings, which have been detected in the surface or the interior of amorphous and crystalline silica [39,40].”
[35] Mosallanejad, S., Dlugogorski, B.Z., Kennedy, E.M., Stockenhuber, M., Lomnicki, S.M., Assaf, N.W. and Altarawneh, M. Formation of PCDD/Fs in oxidation of 2-chlorophenol on neat silica surface. Environ. Sci. Tech. 2016, 50(3), 1412−1418.
[37] Pan, W.; Zhong, W.; Zhang, D.; Liu, C. Theoretical study of the reactions of 2-chlorophenol over the dehydrated and hydroxylated silica clusters. J. Phys. Chem. A 2012, 116, 430−436.
[39] Zhang, D.J. and Zhang, R.Q. A structural model of one-dimensional thin silica nanowires. Chem. Phys. Lett. 2004, 394(4-6), 437−440.
[40] Hosono, H., Ikuta, Y., Kinoshita, T., Kajihara, K. and Hirano, M. Physical disorder and optical properties in the vacuum ultraviolet region of amorphous SiO2. Phys. Rev. Lett. 2001, 87(17), 175501.

Reviewer 4 Report
Comments and Suggestions for Authors
In the current paper the authors report a computational study on formation of pre-PCTA/DT intermediates from 2-chlorothio-phenol on different silica clusters. The insights derived from this work could have a substantial impact on understanding the PCTA/DTs reactivity and their behaviors towards silica materials. However, a few more details could improve the manuscript to reach publication standard. Please see the comments below:
1. My major concern is about the entropy when two or more 2-CTPs interacting with silica clusters. I recommend to report or comment on the Gibbs free energy, why authors didn’t include the entropy effect?
2. Authors discussed the 2-CTPs interactions with both the silica clusters in detail, however they failed to discuss the results e.g., Figure 2b energy profile why pathway 3 is more favorable than other two (pathways 1 and 2). Why IM6 is more than two-fold stable compared to IM2 and Im4? Is there any favorable interaction between 2-CTPs Therefore, I would recommend validating these calculations at least single point calculations with higher functional (like double hybrid B2PLYP or other) before drawing any conclusions.
3. Also please provide the reason why hydroxylated silica dissociates S-H bond slowly compared to normal silica?
4. I am just curious whether is there any additional interaction between phenyl chloride and silica cluster? Also, what is the barrier for Si-H dissociation to O2 rather than O1?
Based on everything mentioned above, I would recommend publication after minor revision.
Comments on the Quality of English LanguageNA
Round 2
Reviewer 3 Report
Comments and Suggestions for Authors
I have no more comments that would necessitate another review cycle.